# Association of Polymorphic Variants in Argonaute Genes with Depression Risk in a Polish Population

**DOI:** 10.3390/ijms231810586

**Published:** 2022-09-13

**Authors:** Mateusz Kowalczyk, Edward Kowalczyk, Grzegorz Galita, Ireneusz Majsterek, Monika Talarowska, Tomasz Popławski, Paweł Kwiatkowski, Anna Lichota, Monika Sienkiewicz

**Affiliations:** 1Babinski Memorial Hospital, Aleksandrowska St. 159, 91-229 Lodz, Poland; 2Department of Pharmacology and Toxicology, Medical University of Lodz, Zeligowskiego St. 7/9, 90-752 Lodz, Poland; 3Department of Clinical Chemistry and Biochemistry, Medical University of Lodz, Mazowiecka 5, 92-215 Lodz, Poland; 4Department of Clinical Psychology and Psychopathology, Institute of Psychology, University of Lodz, Smugowa St. 10/12, 91-433 Lodz, Poland; 5Department of Microbiology and Pharmaceutical Biochemistry, Medical University of Lodz, Mazowiecka 5, 92-215 Lodz, Poland; 6Department of Diagnostic Immunology, Pomeranian Medical University in Szczecin, Powstancow Wielkopolskich Av. 72, 70-111 Szczecin, Poland; 7Department of Pharmaceutical Microbiology and Microbiological Diagnostic, Medical University of Lodz, Muszynskiego St. 1, 90-151 Lodz, Poland

**Keywords:** argonaute, AGO, polymorphism, depression

## Abstract

Argonaute (AGO) proteins, through their key role in the regulation of gene expression, participate in many biological processes, including cell differentiation, proliferation, death and DNA repair. Accurate regulation of gene expression appears to be important for the proper development of complex neural circuits. Loss of AGO proteins is known to lead to early embryonic mortality in mice with various malformations, including anomalies of the central nervous system. Single-nucleotide polymorphisms (SNPs) of AGO genes can lead to deregulation of the processes in which AGO proteins are involved. The contribution of different SNPs in depression has been extensively studied. However, there are hardly any studies on the contribution of AGO genes. The aim of our research was to assess the relationship between the occurrence of depression and the presence of SNPs in genes AGO1 (rs636882) and AGO2 (rs4961280; rs2292779; rs2977490) in a Polish population. One hundred and one subjects in the study group were diagnosed with recurrent depressive disorder by a psychiatrist. The control group comprised 117 healthy subjects. Study participants performed the HDRS (Hamilton Depression Scale) test to confirm or exclude depression and assess severity. The frequency of polymorphic variants of genes AGO1 (rs636882) and AGO2 (rs4961280; rs2292779; rs2977490) was determined using TaqMan SNP genotyping assays and the TaqMan universal PCR master mix, no AmpErase UNG. The rs4961280/AGO2 polymorphism was associated with a decrease in depression occurrence in the codominant (OR = 0.51, *p* = 0.034), dominant (OR = 0.49, *p* = 0.01), and overdominant (OR = 0.58, *p* = 0.049) models. Based on the obtained results, we found that the studied patients demonstrated a lower risk of depression with the presence of the polymorphic variant of the rs4961280/AGO2 gene—genotype C/A and C/A-A/A.

## 1. Introduction

Depression is the most common mental illness and the main cause of disability and suicide attempts [1,2]. The pathogenesis of depression is still unknown, although some theories have been proposed and recognized, e.g., the monoamine hypothesis, changes in the hypothalamic–pituitary–adrenal axis, inflammation of the nervous system, neuroplasticity and epigenetics [3,4,5]. 

Cognitive deficits, mainly in the area of attention processes, are an important element of the clinical picture in recurrent depressive disorders [6,7]. They build up gradually but relatively quickly, which differentiates them from the clinical picture of dementia [8]. In this group, there are often atrophic and vascular structural changes in the brain, mainly in the area of the frontal and temporal lobes (hippocampus and amygdala) [9]. Studies show that in patients with depression, the following is observed: a reduction in the volume of the frontal lobes, including the prefrontal cortex, and a a reduction in the volume of the limbic structures, e.g., gyrus of the hippocampus formation and the amygdala [10]. There is a kind of imbalance between the evolutionarily older limbic system, i.e., the “emotional brain” and the “motivational/regulatory brain”, that is, the frontal lobes. This is also confirmed by numerous neurobiological studies. Neurofunctional tests show increased activity of the amygdala, belonging to the limbic system, and decreased activity of the frontal lobes in response to negative stimuli. The prefrontal cortex in people with depression does not inhibit the activity of subcortical emotional systems located in the evolutionarily older structures of the brain. “Dysfunctional” frontal lobes in patients with depression are not able to meet all the requirements of civilization, which may be of key importance in the etiology of depression [11,12,13]. Additionally, the prefrontal cortex is involved in the fear conditioning process [14,15], and anxiety hyperactivity understood as a fixed personality trait predisposes to the occurrence of depression symptoms [16]. Studies showed that non-coding ribonucleic acids (ncRNAs) play a key role in the pathogenesis of depression (including neuroplasticity and neurogenesis), resulting in significant clinical symptoms (e.g., suicidal behavior) [17,18,19,20]. Considering a large number of ncRNAs, which are reported to be biomarkers of depression, it is difficult to determine which ones are the most clinically relevant. ncRNA is a special type of RNA that is transcribed from DNA but does not encode proteins [21]. It includes micro RNA (miRNA), round RNA (circRNA), long non-coding RNA (lncRNA) and other, as yet, undiscovered small RNA [22]. miRNA and small interfering RNA (siRNA) represent a group of small regulatory RNAs (srRNAs). These are molecules ranging from ~20 to ~30 nucleotides (nt) in length [23]. Together with Argonaute (AGO) proteins, they form RNA-induced silencing complex (RISC), or central effector complex, which is involved in the process of RNA interference (RNAi), being an evolutionary conservative mechanism of gene expression silencing, initiated by double-stranded RNA (dsRNA) [24,25,26].

Within the RISC complex, srRNAs act as specific probes that enable their protein partners to recognize complementary transcripts or regions of DNA which are subject to regulation. Mature miRNAs and siRNAs act as guides for the AGO protein in search for complementary target sequences present in the transcripts. AGO proteins are responsible for initiating a cascade of signals aimed at silencing gene expression through translation repression, deadenylation and mRNA degradation. This phenomenon is called post-transcriptional gene silencing (PTGS) [27,28,29].

According to Capitão et al. [30], AGO-mediated post-transcriptional silencing can occur through cleavage or translational repression of target mRNAs, while transcriptional silencing can be controlled by DNA methylation and chromatin remodeling. AGO proteins, through their key role in the regulation of gene expression, participate in many biological processes, including cell differentiation, proliferation, death and DNA repair [31]. The precise regulation of gene expression through the RNA interference pathway seems to be extremely important for proper development and maintenance of complex neural circuits [32,33].

Loss of AGO2 leads to early embryonic mortality in mice with various malformations, including anomalies of the central nervous system. AGO2-deficient embryos are delayed in development, show severe phenotypic defects such as heart failure and impaired neural tube closure [34,35]. Single-nucleotide polymorphisms (SNPs) of AGO genes can lead to global deregulation of the processes in which AGOs are involved. Although the contribution of different SNPs in depression has been extensively studied [36,37,38], there are hardly any studies on genes in the miRNA biogenesis pathway, including AGO genes. Thus, the aim of the author’s own research was to assess the relationship between the occurrence of depression and the presence of SNP in genes: AGO1 (rs636882) and AGO2 (rs4961280; rs2292779; rs2977490). Our research is the first to analyze the relationship of these genes with the occurrence of depression in a Polish population.

## 2. Results

### 2.1. Characteristics of the Studied Groups

The mean age of all subjects (*n* = 218) was 39.79 years, standard deviation (SD)—14.02, minimum age—19 years, and maximum age—81 years. Female sex predominated in both groups with recurrent depressive disorders and in the control group (over 70% of participants in the control group and 80% in the group with recurrent depressive disorders—Figure 1). The group with depressive disorders demonstrated most severe (21%) and moderate (63%) degrees of disease symptoms according to the Hamilton Depression Scale (HDRS; Figure 2). Genotyping was successfully performed for all samples in this study. Allele and genotype frequencies among the patients and controls were in the Hardy–Weinberg equilibrium (HWE) except for the rs229779/AGO2 (Table 1, *p* < 0.05). The rs4961280/AGO2 polymorphism was associated with a decrease in depression occurrence in the codominant (OR = 0.51, *p* = 0.034), dominant (OR = 0.49, *p* = 0.01), and over-dominant (OR = 0.58, *p* = 0.049) models. There were no differences in the genotype distributions between depression patients and controls for the other studied polymorphisms of AGO1 and AGO2 (Table 2). 

### 2.2. Analysis of a Relationship between the Occurrence of Depression and the Studied Polymorphic Variants of AGO Genes

This assessment was carried out with the use of association studies. These are population studies that allow to determine whether a specific allele of a given gene is more common in a group of unrelated affected individuals than in healthy individuals. For each polymorphism, the frequency of individual genotypes is presented in relation to the presence or the absence of depression. Four genetic models were analyzed: codominant, dominant, recessive and overdominant. The codominant model is the most general model which assumes that each genotype generates a distinct, independent risk. This model compares heterozygous and homozygous genotypes for the variant allele with homozygous ones for the most common allele. The dominant and recessive models assume that the variant allele is sufficient to increase the risk of depression (one for the dominant model and two for the recessive model). The last model, an overdominant one, assumes that only heterozygote contributes to the risk of depression.

Table 2 shows the correlation of depression with the frequency of genotypes of the rs636882/AGO, rs2292779/AGO2, rs2977490/AGO2, and rs4961280/AGO2 polymorphism genotypes. The correlation between the codominant, dominant, and overdominant models and depression was only observed in rs4961280/AGO2 polymorphism.

## 3. Discussion

Genetic variations, such as SNPs in miRNAs or at miRNA binding sites, can influence miRNA-dependent regulation of gene expression, which is associated with a variety of diseases, including depression, and can alter individual susceptibility to the disease. In our work, we looked for a relationship between AGO1 and AGO2 polymorphisms and the risk of depression. The studied population, appeared to demonstrate is a lower risk of depression for the polymorphic variant of the rs4961280/AGO2 gene—genotype C/A and C/A-A/A. We are aware that genetic polymorphisms rarely determine the development of a disease. We would rather point out modification (increase or decrease) of susceptibility or resistance to factors contributing to the occurrence of the disease and/or the severity of its course. This is due to the fact that the phenotype depends not only and directly on the genotype, but also on its interaction with the environment. We should also bear in mind that interactions between genes and the environment are very complex. The body is exposed to a number of both positive and negative factors [39]. Further, many genes show pleiotropic properties, which means that they are responsible for the development of many phenotypic features, and not only for one feature, as it was believed according to the classic, already outdated definition of a gene. Despite concerns that a particular gene polymorphism may be linked to a particular disease, numerous studies on these relationships have been carried out. He et al. [40] investigated a relationship between genetic variants of miRNA processing genes and depression. They analyzed DGCR8, AGO1 and GEMIN4 genes 314 patients and 252 healthy subjects participated in this study. There was a significant difference between depressed and healthy patients with regard to DGCR8 rs3757 and AGO1 rs636832 genes. DGCR8 rs3757 was associated with an increased risk of suicidal ideation and a good response to antidepressant therapy, while the AGO1 rs636832 gene was associated with a reduced risk of suicidal behavior. There were no significant differences in GEMIN4 rs7813 between the sick and the healthy. In our own research, we did not look for a relationship between depressive symptoms and the studied polymorphisms. In our study, genotypes C/C, G/C and G/G rs636882/AGO1 were not related to depression. Different results of our research and that conducted by He et al. [40] may result from population differences of the respondents. We have noticed, in turn, that the rs4961280/AGO2 gene plays a protective role against depression.

The main functions of AGO proteins relate to their role in the RNA interference process [41,42]. The course of interference depends to a large extent on the structure of RISC, and especially on the AGO protein, which is the main component necessary for the process to take place. In order to regulate gene expression, the RISC complex targets the appropriate mRNA. This expression takes place in two ways depending on the level of miRNA alignment with mRNA. With perfect complementarity, the RISC complex degrades the mRNA strand, while the non-perfect alignment results in the 5′ end of the miRNA joining the 3′ end of the UTR region, inducing translation repression. Recognition of mRNA by miRNAs is based on sequence complementarity according to several principles: from the 5′ end of miRNA, where the so-called “Region seedsequence” is located [43].

In neurons, translational repression rather than irreversible mRNA degradation is preferred. This helps to stably maintain a pool of specific mRNAs, whose temporarily controlled expression is critical to synaptic development or plasticity [44].

We are not able to explain how the rs4961280/AGO2 polymorphism affects the RISC function, and thus protein synthesis. This problem requires further research. Perhaps not the effect on the synthesis of proteins, but the effect on transport of miRNAs, related to, e.g., with neuronal plasticity, is responsible for the protective role of the rs4961280/AGO2 gene.

Ferreira et al. [45] points out the potential of AGO-2, which serves as a platform providing miRNA. Certain miRNAs act on the same target gene and exert synergetic functions in depression. For example, miRNA-124, miRNA-139-5p and miRNA-34c-5p, targeting the spermidine/spermine N1-acetyltransferase 1 and spermine oxidase gene, regulate neuronal differentiation and proliferation [46]. miRNA-335 and miRNA-1202, by targeting the glutamate receptor metabotropic 4 gene, regulate glutamate metabolism [46,47]. miRNA-124, miRNA-221, miRNA-132, and miRNA-16, by targeting the brain-derived neurotrophic factor gene, regulate synaptic plasticity [48,49,50,51]. miRNA-24-3p and miRNA-425-3p, by targeting MAPK/Wnt-system genes, regulate the MAPK and Wnt signaling pathways [52].

It turns out that not only AGO2 is a miRNA transporter, but also miRNA in complex with the AGO2 protein can participate in the transfer of mRNA from the cytosol to degradation sites, called P-bodies [53,54].

Many researchers believe that dendritic P-bodies and the activity of AGO proteins in dendrites contribute to neuronal plasticity [55,56,57]. Neural plasticity is the basic mechanism of neuronal adaptation that is disturbed in depression. Changes in neuronal plasticity induced by stress and other negative stimuli play an important role in the development of depression. AGO participation in neuronal plasticity is suggested in studies conducted by Patranabis and Bhattacharyya [58]. The authors showed that the activity of the nerve growth factor (NGF—a neurotrophic factor structurally related to the brain-derived neurotrophic factor BDNF) depends on phosphorylation of AGO2 in Tyr529. NGF plays an important role in the functioning of sensory and sympathetic systems, while BDNF is the most common neurotrophic factor studied in the last decade. In mammals, BDNF is responsible for regulation of axon growth and synaptic plasticity [59]. Gershoni-Emek et al. [60] further report that the core of miRISC (AGO2 and miRNA) is located in axon branches and axon growth cones of peripheral nerves, which may support the theory that AGO influences neural plasticity.

Unfortunately, depression is an increasingly common disease, but it is very often difficult to diagnose, especially in basic medical care. Clinicians detect depressive states more easily in younger adults than in older. Depression may be also misdiagnosed mainly of male patients. In contrast, the detection of mild depressive disorders is difficult because symptoms may be confused with those seen under stress. Therefore, rapid tests to assess the risk of depression are necessary to complement the psychological and medical diagnosis. Interdisciplinary research study offers an opportunity to develop a good tool for universal application. It is known that biomarkers such as genetic mutations, neurotransmitters, and cytokines can be used for the identification of depressive disorders. Our research has shown that that SNP rs229779/AGO2 is associated with the risk of depression. It appears that polymorphisms may be potential biomarkers to depression diagnose.

## 4. Materials and Methods

The research was conducted in the period between January 2019 and December 2020. A total of 218 people were examined. All patients in the study group and people in the control group were native, unrelated Caucasian Poles from central Poland. Blood was collected in the study and control groups to evaluate polymorphism of AGO genes. Patients from both groups with chronic inflammatory diseases, neurological diseases, organic disorders, autoimmune diseases, under oncological treatment, metabolically decompensated, with injuries (including head injuries), addicted to psychotropic drugs and narcotics and refusing to consent to participate in this study were excluded.

Each of the respondents expressed their consent in writing to participate in this study in accordance with the protocol approved by the Bioethics Committee of the Medical University of Łódź, No. RNN/402/18 of the European Commission of 10 December 2018. Each person from the control group and the study group also completed the Hamilton Depression Scale form in order to exclude from the control group people with an abnormal result on this scale.

### 4.1. Study and Control Groups

The study group (depressive disorders) consisted of 101 people hospitalized with a diagnosed depressive episode or recurrent depressive disorders at the Department of Adult Psychiatry, Medical University of Lodz and at the Specialist Psychiatric Healthcare Complex in Lodz, Hospital of J. Babiński. All patients were examined upon admission to treatment. The study group included psychiatric patients hospitalized for the first time and not previously treated for depressive disorders, as well as those who had been treated pharmacologically for many years, admitted to the ward to modify their therapy or due to deterioration of health (another affective episode). The sample size was estimated using the G*Power program [61]. One hundred and one subjects in the study group were diagnosed with recurrent depressive disorder (study group, F33) by a psychiatrist according to ICD-10 [62]. The individuals in the study group were matched taking into account the pharmacological treatment applied. Only the patients taking SSRI medications at the time of this study and who were receiving standard treatment with SSRIs were eligible to participate [63]. Study participants performed the HDRS (Hamilton Depression Scale) test to confirm or exclude depression and assess severity. The control group consisted of 117 healthy people with a negative family history of mental illness, matched for age and gender. The individuals for this group were recruited using the snowball method. The respondents also performed HDRS test to exclude people with symptoms of depression. In each case, the mental condition of the respondents was assessed by a psychiatrist. 

In the study group, subjects diagnosed with disorders other than study group (F33) were excluded from this study (axis I and axis II), together with subjects in both groups who had confirmed central nervous system damage, neurological diseases, neoplastic diseases and other disease entities in the history, which could significantly affect their cognitive performance. The characteristic of the study and control groups is presented in Table 3.

### 4.2. Molecular Methods

#### 4.2.1. DNA Isolation

DNA for genotyping was isolated from blood. Blood samples were collected in EDTA tubes as anticoagulant (Sarstedt, Nümbrecht, Germany). The QIAamp DNA Blood Mini Kit (Qiagen, Chatsworth, CA, USA) was used for DNA isolation according to the protocol included in the kit provided by the manufacturer. After isolation, DNA samples were stored at −20 °C in TE buffer (pH 8.0). DNA concentration and purity of DNA preparations were determined spectrophotometrically by measuring absorbance at 260 and 280 nm on a nanodrop microspectrofluorimeter (Thermo Scientific, Waltham, MA, USA).

#### 4.2.2. Determination of Single-Nucleotide Polymorphisms (SNPs)

The frequency of polymorphic variants of genes AGO1 (rs636882) and AGO2 (rs4961280; rs2292779; rs2977490) was determined using TaqMan^®^ SNP Genotyping Assays and the TaqMan Universal PCR Master Mix, No AmpErase UNG (Applied Biosystems, Foster City, CA, USA). A kit with primers and fluorescently labeled molecular probes was used to read the genotype during real-time DNA polymerase chain reaction analysis. The markings were made in accordance with recommendations attached by the manufacturer. The reaction was performed on a Stratagene Mx3005p system (Agilent Technologies, Santa Clara, CA, USA). For each polymorphism, the frequency of individual genotypes is presented in relation to the presence or the absence of depression. Four genetic models were analyzed: codominant, dominant, recessive and overdominant. The codominant model is the most general model which assumes that each genotype generates a distinct, independent risk. This model compares heterozygous and homozygous genotypes for the variant allele with homozygous ones for the most common allele. The dominant and recessive models assume that the variant allele is sufficient to increase the risk of depression (one for the dominant model and two for the recessive model). The last model, an overdominant one, assumes that only heterozygote contributes to the risk of depression.

#### 4.2.3. Statistical Analysis

A statistical analysis was performed with STATISTICA 6.0 software (StatSoft Inc., Tulsa, OK, USA). The consistency of the genotype distribution with the Hardy–Weinberg (HWE) distribution was checked using the chi-square (χ^2^) test. The χ^2^ test was used to assess the significance of differences between frequencies of genotypes in the participants from both the groups. Results for which *p* < 0.05 were considered statistically significant. The statistical analysis also determined the risk of an event (odds ratio—OR) and the confidence interval (95% CI) with the use of a linear regression model.

## 5. Conclusions

At present, we cannot clearly identify mechanisms linking the polymorphism of the rs4961280/AGO2 gene with depression. Thus, we still do not fully understand precise mechanisms by which AGO2 works in depression. However, in recent years, tremendous progress has been made, i.e., AGOs have been discovered and their functions have been described. More research is needed so as to confirm that AGO2 plays a key role in the pathogenesis of depression and to provide additional evidence that the regulation of AGO2 expression or activation can be used as a tool in prevention of depression. Nevertheless, our findings suggest that polymorphisms could be potential biomarkers for diagnosing and assessing the risk of depression.

The fact that the distribution of genotypes of the analyzed rs229779/AGO2 polymorphism was inconsistent with the HWE condition also requires clarification. HWE is an important, fundamental principle of population genetics, according to which population genotype frequencies remain constant from generation to generation in the absence of external disturbance. There are several reasons for the HWE imbalance—errors in genotyping are commonly suggested, especially those occurring in case–control association studies. This assumption is based on the proposition that in a large, randomly mating population, genotype frequencies should be consistent with the HWE equilibrium. Nevertheless, it cannot be a ground for unequivocal rejection of the study results due to alleged, suggested genotyping errors. First, there are usually too few genotyping errors and they do not deviate from the Hardy–Weinberg equilibrium. Moreover, a number of heterozygotes is an important parameter suggesting a significant contribution of genotyping errors to deviation from the HWE equilibrium. The reduced frequency of heterozygosity (LoH) is indicative of factors responsible for deviations from the HWE equilibrium other than genotyping errors, including clean-up selection, copy number variation, inbreeding, and population substructure. An additional analysis verifying the correctness of genotyping is recommended in case of an excess of heterozygotes (GoH). The analysis of the frequency of heterozygosity performed in our study does not indicate the occurrence of the GoH phenomenon, and thus does not indicate any errors in genotyping. Hence, our conclusions have a strong scientific basis and facilitate understanding the genetic basis of depression. We believe that SNP rs229779/AGO2 is associated with the risk of depression. Nevertheless, so as to provide stronger evidence, it is advisable to confirm our observations using a larger group of patients with depression.

### Limitations and Future Directions

Nevertheless, so as to provide stronger evidence, it is advisable to confirm our observations using a larger group of patients with depression taking into account both sexes and gender-related relationships of depression risk in the study group.

## Figures and Tables

**Figure 1 ijms-23-10586-f001:**
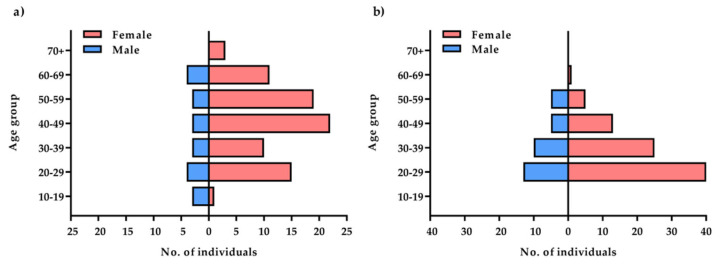
Comparison of the study: (**a**) and control (**b**) groups by age and gender.

**Figure 2 ijms-23-10586-f002:**
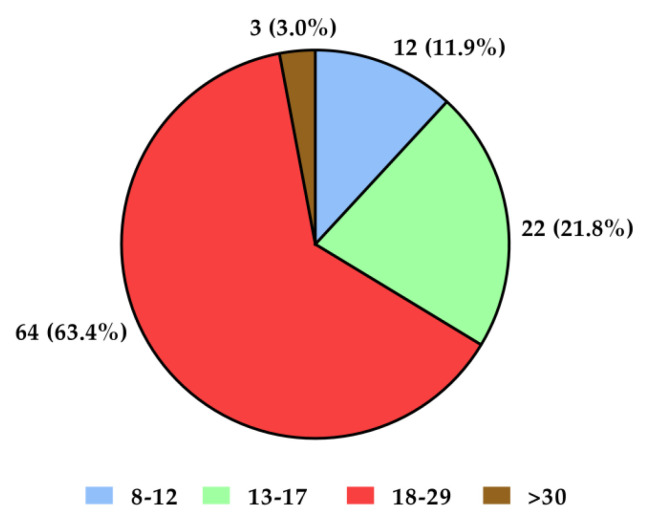
Severity of depression symptoms in the study group according to the HDRS (Hamilton Depression Scale) test. Interpretation of results: <7—no symptoms of depressive disorders; 8–12—mild severity; 13–17—moderate severity; 18–29—severe severity; >30—very severe intensification of depressive disorders symptoms.

**Table 1 ijms-23-10586-t001:** Concordance of genotype distribution of the analyzed miRNA gene maturation.

Polymorphism/Gene	*p*-Value *		
	Totality	Control Group	Study Group
rs636882/AGO1	0.15	0.45	0.11
rs2292779/AGO2	0.00071	0. 0097	0.03
rs2977490/AGO2	0.87	0.66	0.36
rs4961280/AGO2	0.87	0.84	0.59

* *p* < 0.05—in accordance with the Hardy–Weinberg equilibrium.

**Table 2 ijms-23-10586-t002:** Correlation of depression with the frequency of genotypes of the rs636882/AGO1, rs2292779/AGO2, rs2977490/AGO2, and rs4961280/AGO2 polymorphism genotypes.

Polymorphism/Gene	Model	Genotype	Control Group	Study Group	OR (95% CI)	*p*-Value *
rs636882/AGO1	Codominant	G/G	87 (75%)	77 (75.5%)	1.00	0.82
		G/C	26 (22.4%)	21 (20.6%)	0.91 (0.48–1.75)	
		C/C	3 (2.6%)	4 (3.9%)	1.51 (0.33–6.94)	
	Dominant	G/G	87 (75%)	77 (75.5%)	1.00	0.93
		G/C-C/C	29 (25%)	25 (24.5%)	0.97 (0.53–1.80)	
	Recessive	G/G-G/C	113 (97.4%)	98 (96.1%)	1.00	0.58
		C/C	3 (2.6%)	4 (3.9%)	1.54 (0.34–7.04)	
	Overdominant	G/G-C/C	90 (77.6%)	81 (79.4%)	1.00	0.74
		G/C	26 (22.4%)	21 (20.6%)	0.90 (0.47–1.72)	
rs2292779/AGO2	Codominant	G/G	39 (33.6%)	30 (29.4%)	1.00	0.79
		C/G	44 (37.9%)	40 (39.2%)	1.18 (0.62–2.24)	
		C/C	33 (28.4%)	32 (31.4%)	1.26 (0.64–2.49)	
	Dominant	C/G	39 (33.6%)	30 (29.4%)	1.00	0.5
		C/G-C/C	77 (66.4%)	72 (70.6%)	1.22 (0.68–2.16)	
	Recessive	G/G-C/G	83 (71.5%)	70 (68.6%)	1.00	0.64
		C/C	33 (28.4%)	32 (31.4%)	1.15 (0.64–2.06)	
	Overdominant	G/G-C/C	72 (62.1%)	62 (60.8%)	1.00	0.85
		C/G	44 (37.9%)	40 (39.2%)	1.06 (0.61–1.82)	
rs2977490/AGO2	Codominant	G/G	55 (47.4%)	49 (48%)	1.00	0.42
		G/A	52 (44.8%)	40 (39.2%)	0.68 (0.49–1.52)	
		A/A	9 (7.8%)	13 (12.8%)	1.62 (0.64–4.12)	
	Dominant	G/G	55 (47.4%)	49 (48%)	1.00	0.93
		G/A-A/A	61 (52.6%)	53 (52%)	0.98 (0.57–1.66)	
	Recessive	G/G-G/A	107 (92.2%)	89 (87.2%)	1.00	0.22
		A/A	9 (7.8%)	13 (12.8%)	1.74 (0.71–4.25)	
	Overdominant	G/G-A/A	64 (55.2%)	62 (60.8%)	1.00	0.4
		G/A	52 (44.8%)	40 (39.2%)	0.79 (0.46–1.36)	
rs4961280/AGO2	Codominant	C/C	48 (41.4%)	60 (58.4%)	1.00	0.034
		C/A	55 (47.4%)	35 (34.3%)	0.51 (0.29–0.90)	
		A/A	13 (11.2%)	7 (6.9%)	0.43 (0.16–1.16)	
	Dominant	C/C	48 (41.4%)	60 (58.8%)	1.00	0.01
		C/A-A/A	68 (58.6%)	42 (41.2%)	0.49 (0.25–0.85)	
	Recessive	C/C-C/A	103 (88.8%)	95 (93.1%)	1.00	0.26
		A/A	13 (11.2%)	7 (6.9%)	0.58 (0.22–1.53)	
	Overdominant	C/C-A/A	61 (52.6%)	67 (65.7%)	1.00	0.049
		C/A	55 (47.4%)	35 (34.3%)	0.58 (0.34–1.00)	

* *p* < 0.05.

**Table 3 ijms-23-10586-t003:** The demographic characteristic of the study and control groups.

Groups	Variables		
	Age (Years) Mean ± SD	Sex *n* (%)	
		Female	Male
**Study group** **(*n* = 101)**	44.3 ± 19.1	81 (80.2)	20 (19.8)
** Control group** **(*n* = 117) **	33.2 ± 9.1	83 (70.9)	34 (29.1)
** All** **(*n* = 218) **	39.8 ± 14.0	164 (75.2)	54 (24.8)

*n*—size; %—percentage; SD—standard deviation.

## Data Availability

Reported results are available from the corresponding author.

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
