# Peer review of "Association of Polymorphic Variants in Argonaute Genes with Depression Risk in a Polish Population"

_ijms, 2022, doi:10.3390/ijms231810586_

Round 1
Reviewer 1 Report
This is most significant empirical research paper which brings more insights into the mechanisms and determinants of depression risk on molecular level. Specific polymorphisms of argonaute genes are reported to have association with greater resilience (or lower risk) for depression.
There are some minor points, which may help to improve the paper.
1. Clarify whether the depressive episode sub-group included patients which prospectively converted into bipolar disorder
2. Add a concluding paragraph of 200 words with a "take home message" from this study for broader audience, including clinicians
3. Perhaps (optionally) consider setting their results in the more advanced context of evidence-based and nomothetic networks psychiatry, e.g.:
https://doi.org/10.1007/s13148-010-0014-2
https://doi.org/10.5498%2Fwjp.v11.i1.1
Author Response
Dear Reviewer 1
Thank you for a helpful review of our manuscript. All your comments helped us to clarify our results and make the article more readable. Your suggestions have been incorporated as appropriate into the revised version of the manuscript. The manuscript has been modified, reorganized and completed. Please find below our answers. We appreciate your contribution that helps improve the manuscript.
Comments and Suggestions for Authors
This is most significant empirical research paper which brings more insights into the mechanisms and determinants of depression risk on molecular level. Specific polymorphisms of argonaute genes are reported to have association with greater resilience (or lower risk) for depression.
There are some minor points, which may help to improve the paper.
1. Clarify whether the depressive episode sub-group included patients which prospectively converted into bipolar disorder.
Answer: This information has been added in the Characteristics of the Studied Groups section.
2. Add a concluding paragraph of 200 words with a "take home message" from this study for broader audience, including clinicians.
Answer: This concluding paragraph has been added.
3. Perhaps (optionally) consider setting their results in the more advanced context of evidence-based and nomothetic networks psychiatry, e.g.:
https://doi.org/10.1007/s13148-010-0014-2
https://doi.org/10.5498%2Fwjp.v11.i1.1
Answer: Thank you for your valuable advice. We will follow the suggestions in our further work.
Thank you for your insightful review and valuable comments.
Reviewer 2 Report
Kowalczyk and colleagues investigated in the present article entitled ‘Association of Polymorphic Variants in Argonaute Genes with Depression Risk in a Polish Population’, the current status of knowledge of contribution of the AGO genes in depression. For this purpose, authors aimed assess the relationship between occurrence of depression and presence of SNPs in genes (AGO1 (rs636882), AGO2 (rs4961280; rs 2292779; rs 2977490)) in Polish people hospitalized with a diagnosis of depressive episode or recurrent depressive disorders. Results showed that the studied patients demonstrated a lower risk of depression with the presence of the polymorphic variant of the rs4961280/AGO2 gene - genotype C/A and C/A-A/A.
The main strength of this communication paper is that it addresses an interesting and timely question, providing evidence for the identification of mechanisms linking the polymorphism of the rs4961280/AGO2 gene with depression. In general, I think the idea of this article is really interesting and the authors’ fascinating observations on this timely topic may be of interest to the readers of International Journal of Molecular Sciences. However, some comments, as well as some crucial evidence that should be included to support the author’s argumentation, needed to be addressed to improve the quality of the manuscript, its adequacy, and its readability prior to the publication in the present form, in particular reshaping parts of the Introduction and Methods sections by adding more evidence and theoretical constructs.
Please consider the following comments:
· Abstract: Please rephrase this section, as data are not presented in an accurate format, and the flow of information is not consistent.
· Introduction: The ‘Introduction’ section is well-written and nicely presented, with a good balance of descriptive text and information about biological biomarkers of depressive disorder. Nevertheless, I believe that more information about pathophysiology and core features of depression will provide a better and more accurate background, because as it stands, this information is not highlighted in the text. In this regard, I would suggest to add more information on pathological neural substrates of depressive disorder, specifically on frontal lobe dysfunction, and on related effects on patients’ cognitive impairments (https://doi.org/10.1111/psyp.14122; https://doi.org/10.3390/biomedicines10040849). In line with this evidence, I would recommend also focusing on description of quantitative assessment with electroencephalography (EEG), as some specific measure, like frontal lobe alpha activity, is highlighted in the quest for physiological correlates of depression (https://doi.org/10.4103/jnrp.jnrp_293_18; https://doi.org/10.3390/biomedicines10081897; https://doi.org/10.1016/j.ebiom.2022.104027).
· Study-Control group: Data about participants and information about clinical assessment for patients’ selection are not adequately explained. For this reason, I would ask the authors to specify inclusion criteria for patients involved in this study, like severity of disorder. Also, could the authors specify how did they estimate the exact number of participants? Did they use a power analysis?
· Results: In my opinion, this section is well organized, but it illustrates findings in an excessively broad way. Authors should provide better describe statistical information, rewriting this section more accurately, to ensure in-depth understanding of their findings.
· Analysis of a Relationship Between the Occurrence of Depression and the Studied Polymorphic Variants of AGO Genes: This paragraph that explains frequency of individual genotypes in relation to the presence or absence of depression is the most important part of the study and should clearly describe all the experimental sessions in detail; therefore, this section might be improved by including further explanations, allowing the effective communication of experimental procedures.
· I would ask the authors to include a proper ‘Limitations and future directions’ section before the end of the manuscript, in which authors can describe in detail and report all the technical issues that could be brought to the surface.
· Figures and Tables: According to the Journal’s guidelines, please add an explanatory caption for each figure/table within the text.
- References: Authors should consider revising the bibliography, as there are several incorrect citations. Indeed, according to the Journal’s guidelines, they should provide the abbreviated journal name in italics, the year of publication in bold, the volume number in italics for all the references.
Overall, the manuscript contains 2 tables, 2 figures and 50 references. I believe that the manuscript may carry important value providing evidence for the identification of mechanisms linking the polymorphism of the rs4961280/AGO2 gene with depression.
I hope that, after these careful revisions, the manuscript can meet the Journal’s high standards for publication. I am available for a new round of revision of this article.
I declare no conflict of interest regarding this manuscript.
Best regards,
Reviewer
Author Response
Reviewer 2
Thank you for a helpful review of our manuscript. All your comments helped us to clarify our results and make the article more readable. Your suggestions have been incorporated as appropriate into the revised version of the manuscript. The manuscript has been modified, reorganized and completed. Please find below our answers. We appreciate your contribution that helps improve the manuscript.
Comments and Suggestions for Authors
Kowalczyk and colleagues investigated in the present article entitled ‘Association of Polymorphic Variants in Argonaute Genes with Depression Risk in a Polish Population’, the current status of knowledge of contribution of the AGO genes in depression. For this purpose, authors aimed assess the relationship between occurrence of depression and presence of SNPs in genes (AGO1 (rs636882), AGO2 (rs4961280; rs 2292779; rs 2977490)) in Polish people hospitalized with a diagnosis of depressive episode or recurrent depressive disorders. Results showed that the studied patients demonstrated a lower risk of depression with the presence of the polymorphic variant of the rs4961280/AGO2 gene - genotype C/A and C/A-A/A.
The main strength of this communication paper is that it addresses an interesting and timely question, providing evidence for the identification of mechanisms linking the polymorphism of the rs4961280/AGO2 gene with depression. In general, I think the idea of this article is really interesting and the authors’ fascinating observations on this timely topic may be of interest to the readers of International Journal of Molecular Sciences. However, some comments, as well as some crucial evidence that should be included to support the author’s argumentation, needed to be addressed to improve the quality of the manuscript, its adequacy, and its readability prior to the publication in the present form, in particular reshaping parts of the Introduction and Methods sections by adding more evidence and theoretical constructs.
Please consider the following comments:
Abstract: Please rephrase this section, as data are not presented in an accurate format, and the flow of information is not consistent.
Answer: Abstract has been corrected.
Introduction: The ‘Introduction’ section is well-written and nicely presented, with a good balance of descriptive text and information about biological biomarkers of depressive disorder. Nevertheless, I believe that more information about pathophysiology and core features of depression will provide a better and more accurate background, because as it stands, this information is not highlighted in the text. In this regard, I would suggest to add more information on pathological neural substrates of depressive disorder, specifically on frontal lobe dysfunction, and on related effects on patients’ cognitive impairments (https://doi.org/10.1111/psyp.14122; https://doi.org/10.3390/biomedicines10040849).
In line with this evidence, I would recommend also focusing on description of quantitative assessment with electroencephalography (EEG), as some specific measure, like frontal lobe alpha activity, is highlighted in the quest for physiological correlates of depression (https://doi.org/10.4103/jnrp.jnrp_293_18; https://doi.org/10.3390/biomedicines10081897; https://doi.org/10.1016/j.ebiom.2022.104027).
Answer: This information has been added in the introduction section.
Study-Control group: Data about participants and information about clinical assessment for patients’ selection are not adequately explained. For this reason, I would ask the authors to specify inclusion criteria for patients involved in this study, like severity of disorder. Also, could the authors specify how did they estimate the exact number of participants? Did they use a power analysis?
Answer: This information has been added in the Study and Control Group section.
Results: In my opinion, this section is well organized, but it illustrates findings in an excessively broad way. Authors should provide better describe statistical information, rewriting this section more accurately, to ensure in-depth understanding of their findings.
Answer: This section was corrected according Reviewer suggestion.
Analysis of a Relationship Between the Occurrence of Depression and the Studied Polymorphic Variants of AGO Genes: This paragraph that explains frequency of individual genotypes in relation to the presence or absence of depression is the most important part of the study and should clearly describe all the experimental sessions in detail; therefore, this section might be improved by including further explanations, allowing the effective communication of experimental procedures.
Answer: This section has been improved according Reviewer suggestion.
I would ask the authors to include a proper ‘Limitations and future directions’ section before the end of the manuscript, in which authors can describe in detail and report all the technical issues that could be brought to the surface.
Answer: The section ‘Limitations and future directions’ has been added according Reviewer suggestion.
Figures and Tables: According to the Journal’s guidelines, please add an explanatory caption for each figure/table within the text.
Answer: It has been added.
References: Authors should consider revising the bibliography, as there are several incorrect citations. Indeed, according to the Journal’s guidelines, they should provide the abbreviated journal name in italics, the year of publication in bold, the volume number in italics for all the references.
Answer: It has been corrected.
Overall, the manuscript contains 2 tables, 2 figures and 50 references. I believe that the manuscript may carry important value providing evidence for the identification of mechanisms linking the polymorphism of the rs4961280/AGO2 gene with depression.
I hope that, after these careful revisions, the manuscript can meet the Journal’s high standards for publication. I am available for a new round of revision of this article.
Thank you for your insightful review and valuable comments.
Round 2
Reviewer 2 Report
In this article Kowalczyk and colleagues explored contribution of the AGO genes in depression disorder.
I only have few last suggestions to do, to further improve the theoretical background of the present paper and its argumentation: in this regard, I would recommend deepening the information pathological neural substrates of depressive disorder, specifically on frontal lobe dysfunction, and on related effects on patients’ cognitive impairments (https://doi.org/10.3109/10253890.2015.1045868; https://doi.org/10.1016/j.tins.2022.04.003; https://doi.org/10.17219/acem/146756).
Overall, this is a timely and needed study, and I look forward to seeing further studies on this issue by these authors in the future.
I am always available for other revisions of such as interesting and important articles.
Thank You for your work.
Author Response
Dear Reviewer,
thank you very much for valuable comments and praise for our work, we improved the article as suggested and included new items of literature.
Sincerely,
Monika Sienkiewicz